# Thermally Enhanced Biodegradation of TCE in Groundwater

Petra Najmanová [1,2,*], Jana Steinová [3,4], Marie Czinnerová [3], Jan Němeček [3,5], Roman Špánek [6], Vladislav Knytl [1,7] and Martin Halecký [2]

1   DEKONTA, a.s., Dřetovice 109, 273 42 Stehelčeves, Czech Republic
2   Department of Biotechnology, University of Chemistry and Technology Prague, Technická 5, 166 28 Prague 6, Czech Republic
3   Institute for Nanomaterials, Advanced Technology and Innovation, Technical University of Liberec, Bendlova 1407/7, 460 01 Liberec, Czech Republic
4   Department of Botany, Faculty of Science, Charles University, Benátská 2, 128 01 Prague, Czech Republic
5   ENACON s.r.o., Na Holém vrchu 708/3, 143 00 Prague 4, Czech Republic
6   Institute of Mechatronics and Computer Engineering, Technical University of Liberec, Bendlova 1407/7, 460 01 Liberec, Czech Republic
7   Institute for Environmental Studies, Faculty of Science, Charles University, Benatská 2, 128 01 Prague 2, Czech Republic
*   Correspondence: najmanova@dekonta.cz

**Abstract:** In situ remediation is usually restricted by temperature, lack of substrate for reductive dechlorination (anaerobic respiration), the presence of dehalogenating microorganisms, and specific bedrock conditions. In this work, trichloroethene (TCE) degradation was studied by a number of methods, from physical–chemical analyses to molecular biological tools. The abundance changes in dechlorinating bacteria were monitored using real-time PCR. The functional genes *vcrA* and *bvcA* as well as the 16S rRNA specific for representatives of genera *Dehalococcoides*, *Dehalobacter*, and *Desulfitobacterium* were monitored. Furthermore, the sulfate-reducing bacteria and denitrifying bacteria were observed by amplifying the functional genes *apsA* and *nirK*. The elevated temperature and the substrate (whey) addition significantly affected TCE dechlorination. The chlorine index decreased after nine weeks from 2.5 to 0.1 at 22 °C, to 1.1 at 17 °C and 1.7 at 12 °C and complete dechlorination was achieved at 22 °C with whey addition. The achieved results of this work show the feasibility and effectiveness of biological dechlorination of TCE enhanced with elevated temperature and whey addition.

**Keywords:** dechlorination; thermal treatment; chlorinated ethenes; TCE; *Dehalococcoides*

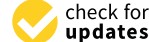

## 1. Introduction

The ubiquitous industrial use of chlorinated solvents caused extensive groundwater contamination. One of the most widespread contaminants is TCE, which is suspected of being a human carcinogen [1]. The traditionally used remediation of contaminated areas was the pump and treat method, but because of its low efficiency and high cost, it has been replaced by in situ technologies based on chemical or biological dehalogenation. Reductive dechlorination is a very effective and promising technology. Its principle is based on a reaction where the chlorinated solvent serves as an electron acceptor and the chlorine atom in its molecule is substituted by the hydrogen atom [2]. Among others, such a process can be performed by various species of bacteria in biological reductive dichlorination, also known as halorespiration [3]. The basic conditions of halorespiration are an anaerobic environment and the presence of an electron donor, hydrogen, which comes from the fermentation of many types of organic substances, natural or purposefully applied to the rock environment [4–6]. The dechlorination rate is affected by geochemical and biological processes and hydrogen availability [7,8]. Biological reductive dechlorination converts stepwise tetrachloroethene to trichloroethene, *cis*-dichloroethene (*cis*-DCE), then to vinyl

chloride (VC), and to ethene [9]. Dechlorination under inappropriate conditions or its low rate can lead to the accumulation of toxic VC or *cis*-1,2-DCE [10]. Due to the much higher toxicity of emerging intermediates, VC is classified as a proven human carcinogen [11], completion of halorespiration up to the final ethene is a very important step.

The rate of biological reductive dechlorination depends on many factors that affect each other such as availability and character of fermenting substrate, temperature, presence of nutrients, the population density of dechlorinating microorganisms, redox potential, or presence of substances used by microorganisms in catabolic processes (e.g., oxygen, nitrates, iron, manganese) [12].

Whereas temperature increase has a positive effect on microbial growth, thermally enhanced biodegradation may accelerate in situ remediation. The influence of temperature on reductive dechlorination was studied under laboratory conditions. These tests showed that the growth of dechlorinating bacteria and connecting dechlorination activity highly depended on temperature [13–15]. Atlas and Bartha (1987) [16] stated that temperature is an important parameter influencing biological reductive dechlorination.

Site remediation technology using reductive dechlorination is a commercially applied technology solving groundwater contamination with chlorinated solvents. Compared to physical–chemical methods, it is a relatively slow method, especially since its start-up often requires a longer time period [15]. For this reason, current research focuses on increasing the efficiency of in situ anaerobic bioremediation by optimizing the temperature conditions with the intention to maximize the rate of biodegradation [13,14]. The higher efficiency of the remediation method will be reflected in the shortening of the time required to reach the target remediation limits and thus in lower remediation costs.

Preliminary screening tests with a wide temperature range of 10–40 °C with individual temperatures tested in 10 °C increments showed temperatures of 20 and 30 °C as most effective for TCE dichlorination [17]. Due to the economy of the process, a lower temperature was finally chosen. Thus, in this study, we have focused on biological remediation of groundwater contaminated with trichloroethene at temperatures 12 °C (as the average temperature of groundwater), 17 °C and 22 °C (as effective and economically feasible temperature for total dechlorination) using whey as a substrate—energy, carbon, and $H^+ + e^-$ sources—for naturally occurring microorganisms, their metabolism providing reductive dechlorination. The dechlorination process was monitored from the chemical as well as biological point of view. TCE content and related products of its degradation were monitored for nine weeks period together with the relative abundance of total biomass and genes specific for enzymes involved in dechlorination, namely *vcrA*, and *bvcA*. Finally, the changes in abundance of three microbial genera contributing to dechlorination (*Dehalococcoides*, *Dehalobacter,* and *Desulfitobacterium*) were monitored using qPCR.

## 2. Materials and Methods

### 2.1. Material Characterization

Groundwater and soil samples were taken at a manufacturing site in Northern Bohemia contaminated by chlorinated volatile organic compounds (cVOCs) originating from historical degreasing activities. An amount of 20 L of groundwater was collected from the injection well AP-2 and 10 kg of soil from the same well was taken during its drilling. The natural groundwater temperature was 12 °C. Immediately after collection, soil and groundwater samples were placed in a refrigerator and during transport, the temperature was maintained between 2–8 °C. Due to the low concentration of cVOCs, it was spiked with TCE (dissolved in methanol) to a TCE concentration of 19.5 mg/L and 145 mg/L TOC (total organic carbon). The results of groundwater analyses after TCE addition used for the laboratory test are in Table 1.

**Table 1.** Concentrations of all parameters of groundwater used for lab tests.

| Parameter | Unit | Concentration at Time 0 |
|---|---|---|
| 1,1-dichloroethene | μg/L | <20 |
| *cis*-1,2-dichloroethene | μg/L | 7430 |
| *trans*-1,2-dichloroethene | μg/L | <20 |
| Trichloroethene | μg/L | 19,450 |
| Tetrachloroethene | μg/L | <20 |
| Vinyl chloride | μg/L | 1310 |
| Σ cVOCs | μg/L | 28,190 |
| Acetylene | μg/L | <1.0 |
| Ethane | μg/L | 10.1 |
| Ethene | μg/L | 119.6 |
| Methane | μg/L | 234.5 |
| TOC | mg/L | 145 |
| Chlorides | mg/L | 98.6 |
| pH | - | 6.48 |
| ORP | mV | 75.0 |
| $Fe^{2+}$ | mg/L | 7.89 |
| $Fe^{3+}$ | mg/L | 0 |

Groundwater contained more than 28 mg/L cVOCs with the highest content of TCE following *cis*-1,2-DCE (Table 1). Groundwater also contained small amounts of toxic vinyl chloride. Furthermore, the water had a relatively high concentration of total organic carbon which increased in the groundwater tested by the addition of methanol, added as a TCE dissolver. The pH value was neutral. Ethene and methane were also detected in groundwater in the lower hundreds of μg/L and about 10 μg/L of ethane.

*2.2. Monitoring*

Only groundwater samples were analyzed during the testing. Soil addition was added due to rock environment simulation on site.

The groundwater samples were analyzed for the wide spectrum of parameters: chlorinated ethenes (VC, 1,1-DCE, *cis*- a *trans*-1,2-DCE, TCE, PCE), ethene, ethane, methane, TOC, pH, redox potential (ORP), temperature, iron, chloride, and relative abundance of bacteria and functional genes using q PCR.

Chlorinated ethenes were evaluated using a gas chromatography–mass spectrometer (GC-MS 6890/5975, Agilent Technologies, Santa Clara, CA, USA) according to ISO 10301:2013. TOC was analyzed according to ISO 1484:1997 using a Liqui TOC II analyzer (Elementar, Langenselbold, Germany). Gases were determined using gas chromatography according to EPA Method RSK-175. The ORP, pH, and temperature were measured with Multi 350i Multimeter (WTW, Weilheim, Germany). The dissolved iron concentration was analyzed using the Spectro Blue inductively coupled plasma atomic emission spectrometer (ICP-AES; SPECTRO, Kleve, Germany) according to ISO 11885:2007. Before analysis groundwater samples were filtered through a 0.45 μm membrane filter. Chlorides were analyzed by titration according to ISO 9297:1989.

Due to the highly volatile chlorinated hydrocarbons, the concentrations of cVOCs and gases were converted to the so-called chlorine number (sometimes referred to as the chloride index; Equation (1)), used to describe the stage of degradation of parent (1) contaminants (TCE and PCE) to degradation products (VC and ethene) according to Bewley et al. (2015) [18]:

$$\text{chlorine number} = \frac{4[\text{PCE}] + 3[\text{TCE}] + 2[\text{DCE}] + 1[\text{VC}] + [\text{ethene}]}{([\text{PCE}] + [\text{TCE}] + [\text{cis} - \text{DCE}] + [\text{VC}] + [\text{ethene}])}$$

where [contaminant] represents the molar concentration (mmol/L) of the individual contaminants.

The molecular methods were the same as described Němeček et al. (2020) [19], see Table 2. qPCR and the detection of the following functional microbial groups were observed: (i) sulfate-reducing bacteria—functional gene *apsA*; (ii) denitrifying bacteria—functional gene *nirK*; dechlorinating bacteria—16S rRNA gene specific for *Dehalococcoides* spp., *Dehalobacter* spp., *Desulfitobacterium* spp.; (iii) functional genes responsible for vinyl chloride reductive dehalogenation (genes *bvcA* and *vcrA*) and (iv) total bacterial biomass expressed by 16S rRNA gene.

**Table 2.** Primers used for q-PCR.

| Primer | Primer Sequences 5′-3′ | Gene | References |
|---|---|---|---|
| Dre441F<br>Dre645R | GTTAGGGAAGAACGGCATCTGT<br>CCTCTCCTGTCCTCAAGCCATA | *Dehalobacter* spp.;<br>gene 16S rRNA | [20] |
| DHC793f<br>DHC946r | GGGAGTATCGACCCTCTCTG<br>CGTTYCCCTTTCRGTTCACT | *Dehalococcoides* spp.;<br>gene 16S rRNA | [21] |
| Dsb406F<br>Dsb619R | GTACGACGAAGGCCTTCGGGT<br>CCCAGGGTTGAGCCCTAGGT | *Desulfitobacterium* spp.;<br>gene 16S rRNA | [20] |
| bvcA277F<br>bvcA523R | TGGGGACCTGTACCTGAAAA<br>CAAGACGCATTGTGGACATC | functional gene of VC reduction,<br>*Dehalococcoides* spp. strain BAV-1 | [22] |
| vcrA880F<br>vcrA1018R | CCCTCCAGATGCTCCCTTTA<br>ATCCCCTCTCCCGTGTAACC | functional gene of VC reduction,<br>*Dehalococcoides* spp. strain VS | [22] |
| RH1-aps-F<br>RH2-aps-R | CGCGAAGACCTKATCTTCGAC<br>ATCATGATCTGCCAGCGGCCGGA | sulfate-reducing bacteria—functional<br>gene *apsA* | [23] |
| nirK876<br>nirK1040 | ATYGGCGGVCAYGGCGA<br>GCCTCGATCAGRTTRTGGTT | denitrifying bacteria—functional<br>gene *nirK* | [24] |
| 16SqPCR-F<br>16SqPCR-R | TCCTACGGGAGGCAGCAGT<br>GGACTACCAGGGTATCTAATCCTGTT | gene for 16S rRNA | [25] |

Total DNA was extracted from groundwater samples. Reactions for qPCR were performed using thermocycler LightCycler® 480 (Roche, Basel, Switzerland). The changes in abundance of representatives of genera *Dehalococcoides*, *Dehalobacter*, and *Desulfitobacterium* were monitored by amplification of 16S rRNA. The genes responsible for reductive dehalogenation (*bvcA* and *vcrA*) were monitored, and primers are given in Table 2.

The results were evaluated by the method of so-called relative quantification, which expresses changes in the amount in relation to a certain starting point—in our case, the input value of the experiment. Therefore, this method allows us to monitor quantitative changes (increases and decreases) in the abundance of individual markers over time.

### 2.3. Lab Test Set-Up

The batch tests were performed under anaerobic conditions at three different temperatures (12, 17, and 22 °C); see Table 3. The three different temperatures were chosen with regard to the natural conditions and to the low cost of its heating on site. Temperature 12 °C simulated the average in situ temperature of groundwater used for lab test. Temperatures 17 °C and 22 °C represented groundwater heating by 5 °C, resp. 10 °C. The dried whey used as an organic substrate was added at a concentration of 1 g/L (100 g of the product containing: 76 g carbohydrates, 13 g proteins, 0.5 g lipids, and 2.8 g NaCl; producer Mogador). The suspension of groundwater and soil in a ratio of 2:1 was tested in two parallel treatments including a blank without substrate addition. All variants were incubated in 250 mL reagent bottles with polypropylene caps in the darkness, unshaken, at an appropriate temperature for nine weeks. All bottles were filled with 100 g of soil (related to the dry matter) and 200 mL of groundwater. Once a week all bottles were hand mixed by circular motion. All analyses were performed only in groundwater at 2, 4, and 9 weeks of the test. The following table describes individual variants of the test.

**Table 3.** Laboratory test set-up.

| Designation of Test Variant | Temperature | Groundwater | Soil (Dry Matter) | Whey Addition |
|---|---|---|---|---|
| 12/W | 12 °C | 200 mL | 100 g | 1 g/L |
| 17/W | 17 °C | 200 mL | 100 g | 1 g/L |
| 22/W | 22 °C | 200 mL | 100 g | 1 g/L |
| 12 | 12 °C | 200 mL | 100 g | - |
| 17 | 17 °C | 200 mL | 100 g | - |
| 22 | 22 °C | 200 mL | 100 g | - |

## 3. Results and Discussion

### 3.1. Chemical Analyses

Figure 1 describes the results of chemical analyses of all chlorinated hydrocarbons (TCE-DCE-VC) and the end products of biological dechlorination (ethane and ethene) in temperature variants with (Figure 1a) and without (Figure 1b) whey for nine weeks. Total dechlorination was observed only in the variant with whey addition at 22 °C after nine weeks. The temperature had a positive effect on dechlorination in this case. On the other hand, tests without whey addition showed the opposite effect of temperature, and the fastest and most effective dechlorination occurred at 12 °C. The explanation may be the stability and survival of the natural microbiome (as shown in Figures 6 and 7) in samples without whey addition adapted at an average of 12 °C detected in the contaminated underground.

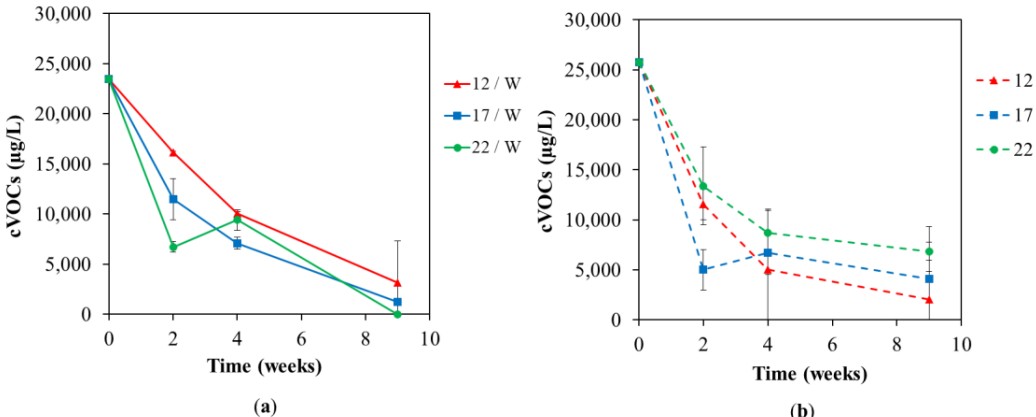

**Figure 1.** The concentration of chlorinated ethenes (cVOCs) in the groundwater at different temperatures of 12 °C, 17 °C, and 22 °C during the lab-scale test with (**a**,**b**) without whey addition.

Previously reported screening tests with the same culture concluded that temperatures of 20 and 30 °C were the most effective for TCE dechlorination [17]. Further reported studies have shown the effect of temperature of partial dechlorination in mixed microbial communities and complete in pure cultures. In these studies, the optimal temperature varies from 22 °C to 65 °C. The highest temperature of 65 °C was reported for thermophilic anaerobic enrichment culture [26]. Reductive dechlorination start-up often requires a longer time period [15]. For this reason, current research focuses on increasing the efficiency of in situ anaerobic bioremediation by optimizing the temperature conditions with the intention of maximizing the rate of biodegradation [13,14]. The specific response of the microbiome on location to temperature must be investigated; however, most commonly reported temperatures are 20–30 °C with deterioration of the process above 35–40 °C [27–32]; thus, thermally enhanced anaerobic dechlorination could be costly, but effective [30]. The highest efficiency of the remediation method will be reflected in the shortening of the time required to reach the target remediation limits through improved biological parameters, e.g., cell growth and (bio)reaction rates [33–35] and physical–chemical, e.g., pollutant desorption,

their volatilization and (bio)availability [36] and releasing of direct electron donors [35] and thus in lower remediation costs. However, costs could be reduced by replacing conventional heating with some modern and sustainable approach, e.g., geothermal heat pumps and solar heating [37] or synergistic coupling dechlorination remediation with underground thermal energy storage [36,38]. However, in this case, the clogging of the pores caused by $Fe^{3+}$ precipitation could also be accompanied by the clogging caused by biomass when additives supporting the decontamination process are added [38].

Most of the partially dechlorinated products (VC and *cis*-1,2-DCE) were at 12 °C (Figure 2). Although the difference from the temperature of 20 °C was not fundamental, the highest temperature used was most suitable for both total dechlorination and prevention of accumulation of partial dechlorinated metabolites. Nonoptimal temperature conditions can lead not only to the deterioration of the dechlorination process and the accumulation of partially dechlorinated products. At suboptimal temperatures (10 and 20 °C), dechlorination of trichloroethene led preferentially to a non-complete process and to the accumulation of VC and *cis*-1,2-DCE [39]. Less effective dechlorination (releasing of less chlorinated by-products) at this temperature was published by other authors [28–30]. Optimal temperature maximizes the rate and shortens the lag phase of dechlorination [29]. Above optimal temperatures finally quickly resulted in a complete loss of degradation activity, but Friis et al. found that not such extreme temperature can also lead to accumulation of *cis*-1,2-DCE from TCE with further partial VC and ethene production at a temperature of 40 °C [39]. Fletcher et al. (2011) stated a similar less effective dechlorination at temperatures of 35 and 40 °C [27]. Thus, a wider spectrum of parameters, e.g., the rate and success of dechlorination, should be monitored and considered to determine the optimal temperature.

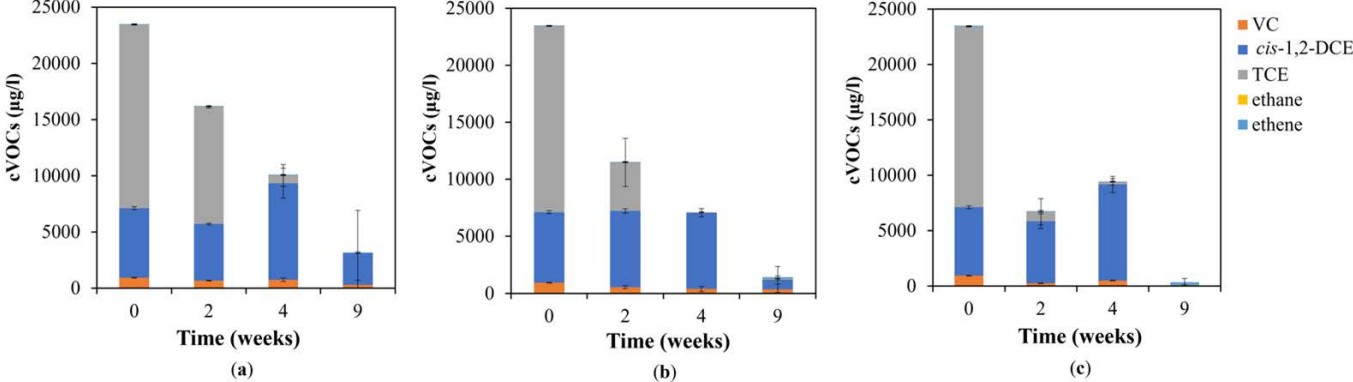

**Figure 2.** Individual concentrations of VC, *cis*-1,2,-DCE, TCE, and dechlorinated products (ethane and ethene) in the groundwater at different lab test temperatures of (**a**) 12 °C, (**b**) 17 °C, and (**c**) 22 °C during the lab-scale test with whey addition.

If we compare different temperatures in the variants with the whey addition, it is obvious that with higher temperatures the faster not only the decrease in TCE but also the increase in intermediates (*cis*-1,2-DCE, VC) and end products was observed (Figure 2). The concentration of end products ethane and ethene was stable in the variants without whey addition and the concentrations of both slightly increased at temperatures of variants of 17 and 22 °C with whey addition (Figure 3a,b). Their concentrations, especially the lack of larger accumulation, indicate that both were utilized by the present microbial culture regardless of the addition of whey.

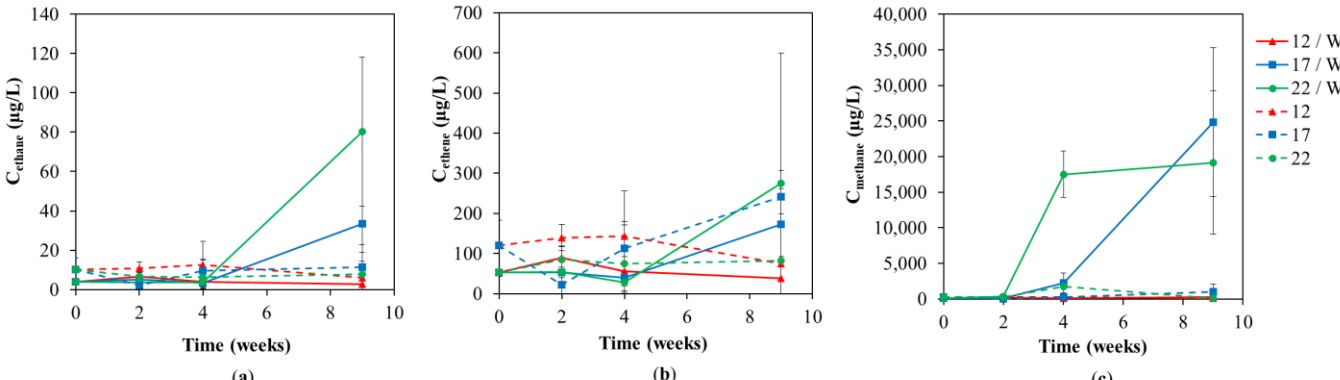

**Figure 3.** Individual concentrations of dechlorinated products (**a**) ethane (**b**) ethene and (**c**) methane in groundwater at different lab test temperatures of 12 °C, 17 °C, and 22 °C during the lab-scale test with and without whey addition.

The production of methane was significant only in variants with the addition of whey at temperatures of 17 °C and 22 °C (Figure 3c). Although some methanogenic microorganisms can grow and digest at temperatures below 10 °C, the methanogens in general are typical thermophilic or mesophilic [40]. Hence, the production of methane only in the variants exposed to elevated temperatures well agrees with the reported optimum temperatures for this group of microorganisms. Specifically, the strongest production of methane was observed after four weeks (17,500 ± 3252 µg/L) at 22 °C while at 17 °C after nine weeks (24,800 ± 10,465 µg/L). Furthermore, the same temporary increase in methane concentration (by one order of magnitude; from 280 ± 260 to 1755 ± 2128 µg/L) induced by temperatures of 22 °C was also observed without whey addition. This delay in methane production (usually 2–3 weeks for this type of remediation [41]) is typical because methane (and reductive dechlorination as well) must precede (syntrophic) fermentation of carbon substrates to produce short fatty acids and $H_2$ [42]. Friis et al. (2007) published that at higher temperatures of 30–40 °C, methanogenesis became absolutely predominant above reductive dechlorination when dechlorinating bacteria not very successfully competed for limited sources, e.g., carbon substrates and/or reducing equivalents (no substrate added) [39,43].

In this study, whey was applied as an accelerating carbon and energy source. In many cases, pure chemicals are used. The most typical in research studies is lactate, which is a by-product of the fermentation process and can also serve as an effective electron donor and organic carbon source [7,14,44–46]. However, many other sources were successfully tested, e.g., emulsified vegetable oils [47,48], glucose [47,49], acetate [50], formate, and fumarate [51]. Additionally improved, multi-purpose (containing also sources of surfactants and vitamins and/or pH control agents) and slow-releasing sources were tested [52,53]. Whey [17,54] and molasses [55] as by-products of the food industry can be an effective way to intensify by sources of electron donors and organic carbon. Because those substances are by-products, the price of the product is limited to transport. The above-mentioned studies demonstrate that those by-products can be effective bioremediation substrates at a cost that is orders of magnitude lower than that of other frequently used alternatives.

Another argument for considering intensification of dechlorination via an electron donor and organic carbon source is the long-term study of in situ bioremediation by Schaefer et al. (2018) even matched a long-term impact of lactate addition when despite the absence of lactate (after complete utilization), the biogeochemical conditions established by lactate addition remained favorable for reductive dechlorination [56].

The pH value was in the neutral pH range of 6.5–7.1 throughout the test in all temperature variants without whey. Regarding the variants with the addition of whey, after its dissolution, it was acidified to values approaching 5.5 due to dissolved whey proteins [57], the release of amino acids through enzymatic hydrolysis, and finally, their fermentation

produces short fatty acids [42]. It rose again to the neutral range due to the depletion (utilization) of the organic substrate at the end of the test at 17 and 22 °C, which corresponds to TOC concentrations (see Figure 4a). The lowest cell activity (fermentation—Figure 5a TOC, methanogenic—Figure 3c and dehalogenation—Figure 5a) at 12 °C led to final pH of 6.05 ± 0.30. However, the range for dechlorinating bacteria is between pH 5.5–8.0 [58–60] and all variants were in this range throughout the test. Therefore, the process was not affected by using whey, which can decrease pH in the above-mentioned ways, and it is not necessary to use a pH control or some additive neutralizing agent [48,61].

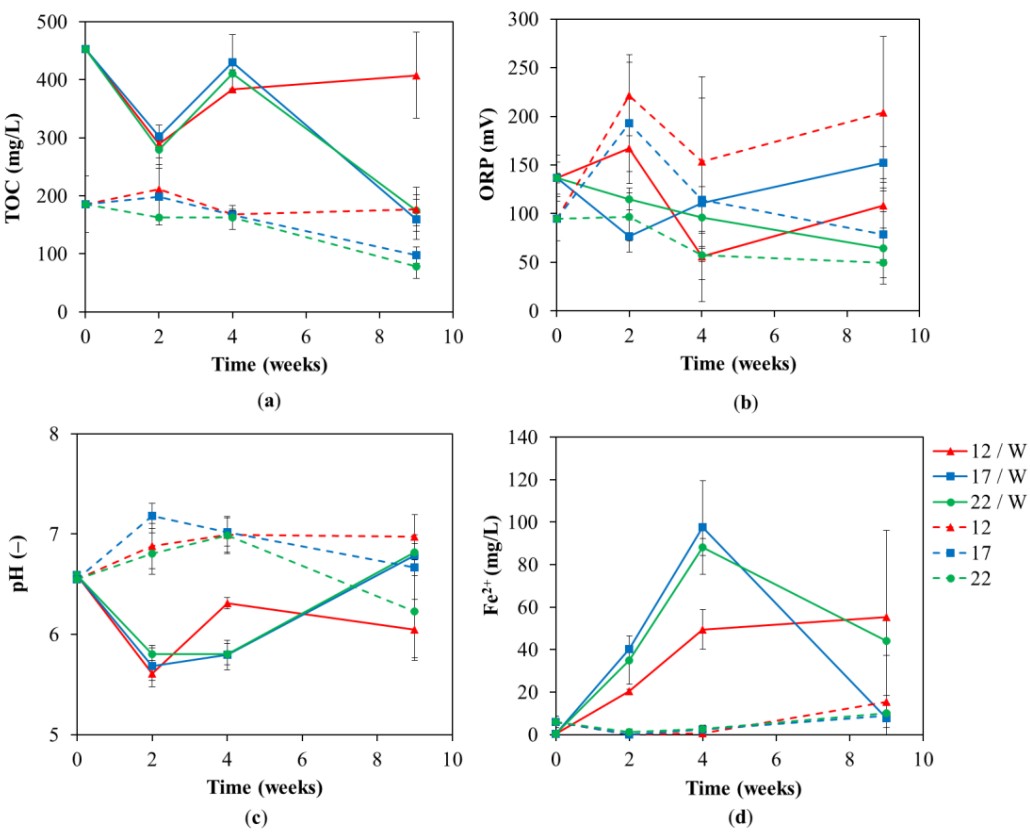

**Figure 4.** (**a**) Total organic carbon (TOC), (**b**) redox potential (ORP), (**c**) pH and (**d**) $Fe^{2+}$ concentration in the groundwater at different lab test temperatures of 12 °C, 17 °C and 22 °C during the lab-scale test with and without whey addition.

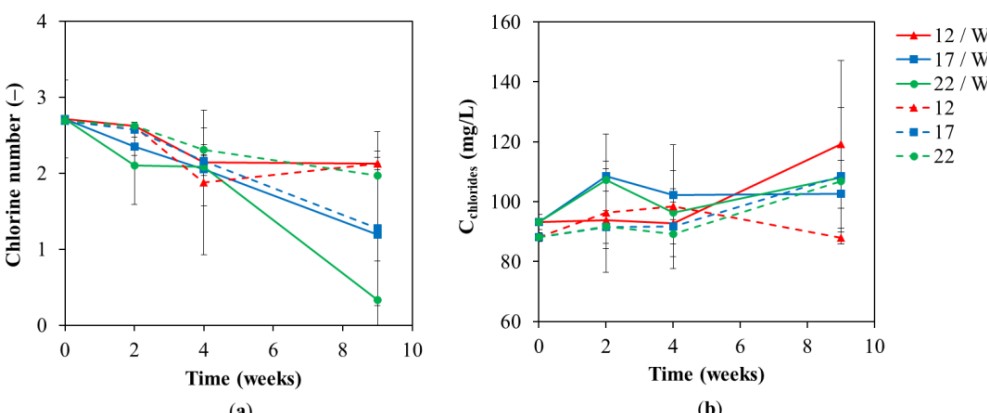

**Figure 5.** (**a**) Chlorine number and (**b**) chlorides concentration in the groundwater at different lab test temperatures of 12 °C, 17 °C, and 22 °C during the lab-scale test with and without whey addition.

TOC increased from 220 to 450 mg/L due to the addition of whey. During the experiment, TOC significantly dropped in the case of variants with whey addition, especially at 22 °C and 17 °C. Even in the case of variants without the whey addition, there was a slight decrease in TOC at 22 and 17 °C (see Figure 4a). At both effective temperatures, 22 °C and 17 °C, the final TOC dropped after nine weeks below the initial natural value of 186 ± 49; therefore, there is no risk of the same accumulation of whey components and contamination of groundwater. The last monitored parameter was the iron content, respectively, its divalent and trivalent forms (Figure 4d). Due to the addition of whey, there was an increase in the reduced form of iron (due to the action of the iron-respiring bacteria [62]) after only two weeks, and its content increased until week four in the following temperature range 17 > 22 > 12 °C. At week 9, the trivalent form rapidly decreased and increased at two higher temperatures. At 12 °C a lower increase in $Fe^{2+}$ was recorded. For variants without the addition of whey, a slight increase in the reduced form was observed at the ninth week. The fundamental increase in the reduced form of iron indicated high activity of iron-respiring bacteria together with methane production (Figure 3c) indicating that the high activity of methanogens can explain some deceleration of reductive dechlorination (Figure 5a, chlorine number) at temperatures of 17 and 22 °C in whey added tests due to competition. Furthermore, analyses of specific bacteria and functional genes also proved the presence and high activity of nitrate- and sulfate-reducing bacteria at temperatures of 17 and 22 °C for whey addition tests. Dechlorinating bacteria are forced to compete in such a complex environment with other microorganisms for energy and carbon sources and especially for electron donors, e.g., $H_2$ [62,63]. The mere availability of an additive carbon source and the simple availability of appropriate final electron acceptors, e.g., sulfates, nitrates, oxidized iron ($Fe^{3+}$), and $CO_2$, can lead to a fundamental suppression of reductive dechlorination [62].

Furthermore, the concentration of chlorides, which are formed during biological reductive dechlorination, was monitored (Figure 5b). The difference between the variants of the tests with and without whey of 6.1 mg/L was caused by the natural presence of chlorides in whey. Chloride accumulation was associated with slow dechlorination kinetics for test variants without whey (Figure 5a,b). For test variants with whey addition, the chloride accumulation in the first two weeks corresponded to the kinetics of dechlorination; the higher temperature, the faster the dechlorination, and the less chloride accumulation. The lack of accumulated chlorides at higher temperatures with whey addition tests can be linked to a fundamental growth of microorganisms in the spectrum of species (Figures 6 and 7) that utilized chlorides as a mineral nutrient.

According to chlorinated ethenes analyses, complete biological reductive dechlorination occurred in the variant with the addition of whey with incubation at 22 °C for nine weeks, which corresponds to the results of Najmanová et al. (2016) [17], in which complete dechlorination occurred in the whey variants at a temperature of 20, respectively, 30 °C. Although the chlorine number was not zero, no chlorinated ethenes were detected in the test variant mentioned above. The value 0.37 (Figure 5a) is based on a specific calculation of the chlorine number, which may not completely coincide with the presence of chlorinated ethenes. Previous studies have described that complete dechlorination usually occurred at temperatures of 15–30 °C [14,30,43,54].

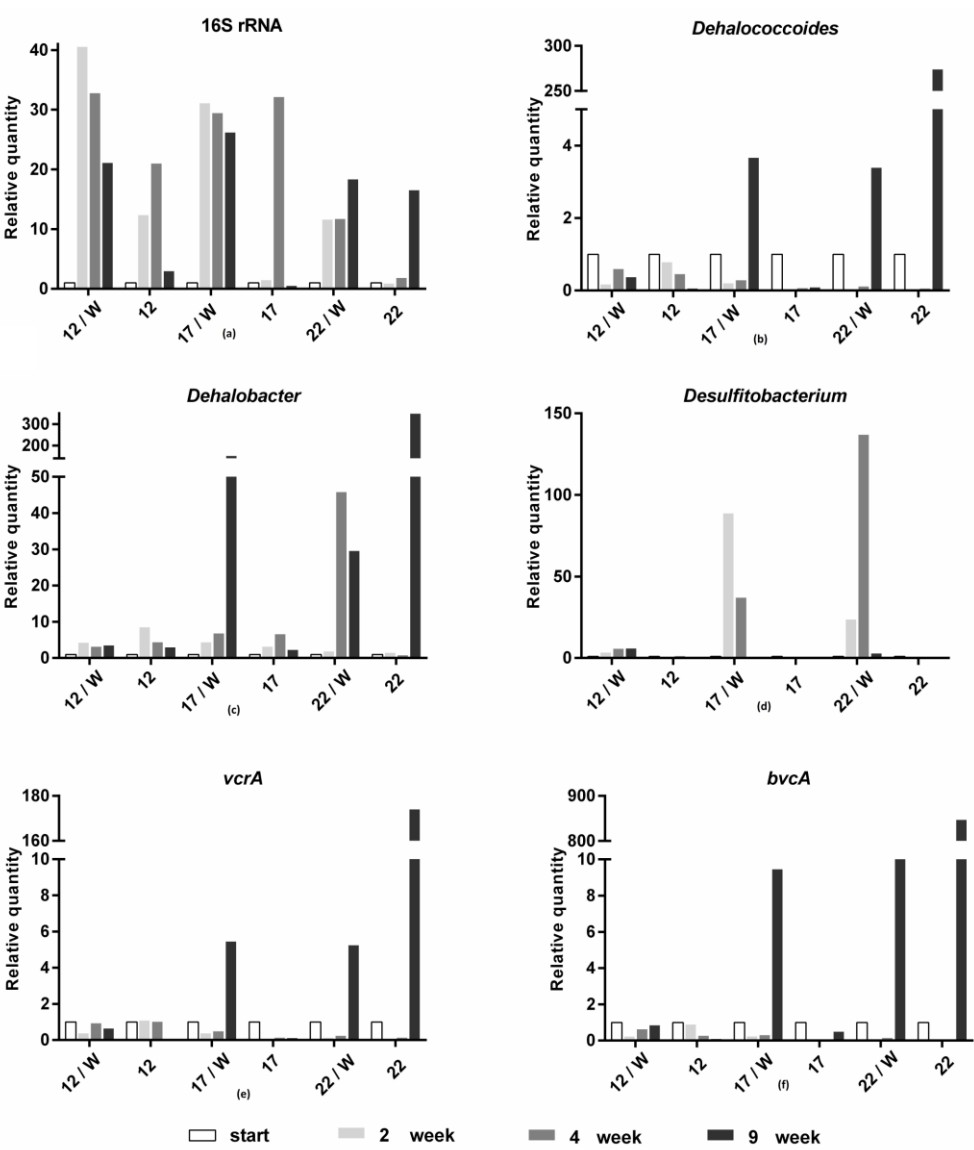

**Figure 6.** Abundance changes in the bacterial biomass (**a**); representatives of dechlorinating bacteria *Dehalococcoides* spp. (**b**), *Dehalobacter* spp. (**c**), and *Desulfitobacterium* spp. (**d**); vinyl chloride reductase genes *vcrA* (**e**) and *bvcA* (**f**) at different temperatures with and without whey addition.

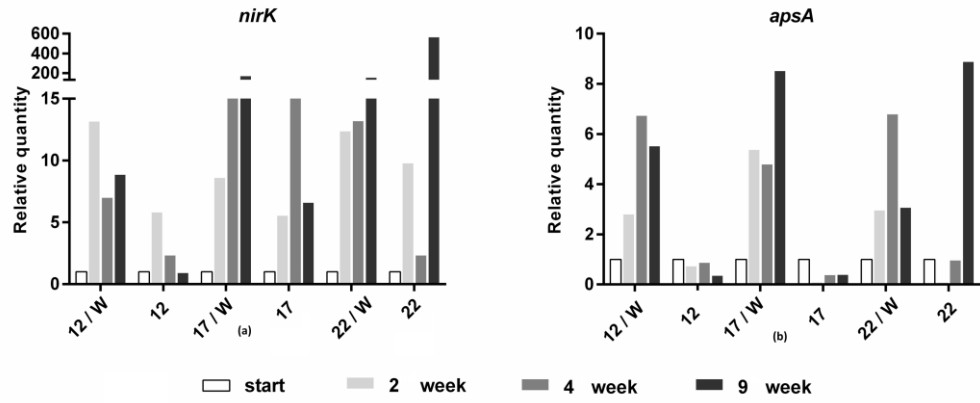

**Figure 7.** Changes in the quantity of (**a**) nitrate- (monitored by *nirK* gene) and (**b**) sulfate- (monitored by *apsA* gene) reducing bacteria in suspensions at different temperatures.

Biological reductive dechlorination requires low ORP values. The largest decrease occurred after two weeks in the 17 °C and 22 °C variants caused by hydrogen production [64] and was associated with a rapid and significant increase in total bacterial biomass (16S rRNA), genera *Dehalobacter* and *Desulfitobacterium* (Figure 6a,c,d) and nitrate- and sulfate-reducing bacteria (Figure 7a,b). Whey-supported reduction of final electron acceptors either original, e.g., nitrates, sulfates, and $Fe^{3+}$, or produced by fermentation ($H_2$) caused a decrease in ORP after two weeks on the order of 22 > 17 > 12 °C, in the case of 22 °C the drop was permanent, while an increase was detected for temperatures of 17 and 12 °C at the end of the tests. Both the pH and ORP time profiles for tests with and without additional substrate are in agreement with a pilot-scale study on chlorinated ethene-polluted groundwater cleanup [52]. In addition, it indicates that the conclusions of our study are applicable to the scale-up of bioremediation technology.

### 3.2. Molecular Biological Analyses

The molecular analysis results are shown in Figures 6 and 7. The overall recovery of bacterial biomass (16S rRNA) increased in all variants, with and without whey, but in the variants with whey, a much greater increase two weeks after the addition of whey was observed. In addition, an increased level of bacterial abundance lasted longer in the whey variants. The greatest increase in bacterial biomass (compared to the initial point) was observed at 12 and 17 °C. For genera *Dehalobacter* and *Desulfitobacterium*, optimal temperatures of 17 and 22 °C have been proven to be optimal. At 12 °C and in the whey-free variants 12 and 17 °C, the amount of the genus *Dehalobacter* increased only slightly in the first half of the experiment and decreased in the later sampling points. In the case of bacteria of the genus *Desulfitobacterium*, we observed that the bacteria belonging to this genus disappeared in the variants without whey. The members of the genus *Dehalococcoides* prospered best at 17 and 22 °C, again with the addition of whey and at 22 °C without whey addition. However, in this case, the increase was observed only after nine weeks. The growth of *Dehalococcoides* corresponded to levels of both functional genes, *vcrA* and *bvcA*, which both followed an identical trend. This finding is consistent with previously published data [28]. However, the abundance of *bvcA* and *vcrA* genes and *Dehalococcoides* must not necessarily always correspond to the degradation efficiency. Marcet et al. (2018) published that the increase in dechlorination efficiency was proportional to the occurrence of *bvcA* and *vcrA* genes as well as representatives of genera *Dehalococcoides* during the temperature shift from 15 to 25 °C, but the decrease in dechlorination efficiency was not directly linked with the decrease in both *bvcA* a *vcrA* genes as well as of genus *Dehalococcoides* [28].

As expected, the relative quantity of *Dehalococcoides* and *vcrA* and *bvcA* genes followed the same trend because the dechlorination ability of members of genus *Dehalococcoides* is ensured by chloroethene reductive dehalogenase genes: *tceA* (transforming TCE to DCE), *vcrA* and/or *bvcA* (transforming *cis*-1,2-DCE to VC and ethene) [47]. Our results confirmed, despite competitive suppression of the abundance of *Dehalococcoides* in the first four weeks (Figure 6b), its essential role in (late phase) dechlorination. Advanced statistics showed that the bacterial dynamic was affected by TCE degradation steps [55].

Members of the genus *Dehalococcoides* are referred to as probably the most widely used pure or predominant member mixture species for the dechlorination process [27,30,46,65] and are found as an important part of natural microbiomes in contaminated sites [10,50,64,66,67]. However, in this study, the genera *Dehalobacter* and *Desulfitobacterium* were detected as more abundant and stable microorganisms involved in reductive dechlorination. Both genera *Dehalobacter* [31,46,65,68] and *Desulfitobacterium* [62,69,70] are referred to as common dechlorinating bacteria in an environment polluted with chlorinated ethenes.

The number of sulfate-reducing bacteria increased after whey addition in all tested temperature variants and only in the variants of 22 °C without whey (Figure 7). Without whey, in contrast, their level decreased at 12 and 17 °C. The same trend was observed for nitrate-reducing bacteria, with the difference that these bacteria grew relatively well even in variants without whey.

The results of molecular biological analyses showed that the largest increase in all markers studied occurred in variants with whey addition incubated at 17 and 22 °C and also in the variant at 22 °C without whey addition. The temperature of 12 °C (simulation of natural site groundwater temperature) is too low for optimal growth of dechlorinating bacteria, although the total bacterial biomass (expressed by 16S rRNA) is growing very well. The addition of whey to support biological reductive dechlorination proved to be beneficial and necessary; in variants without whey, there was no increase in microbial biomass except at a temperature of 22 °C.

The results of the molecular biological analyses corresponded well with the findings of the chemical analyses. A decline of cVOCs in different variants of the experiment, as well as the amount of ethane and ethene produced correlated with the increase in dechlorinating bacteria (see Figure 8). Especially in the fourth week when the chlorine number dropped, *Dehalobacter* started to grow below 17 and 22 °C temperatures in variants with whey addition. The same progress was observed for *Dehalococcoides*, although, in this case, a weaker increase was reported.

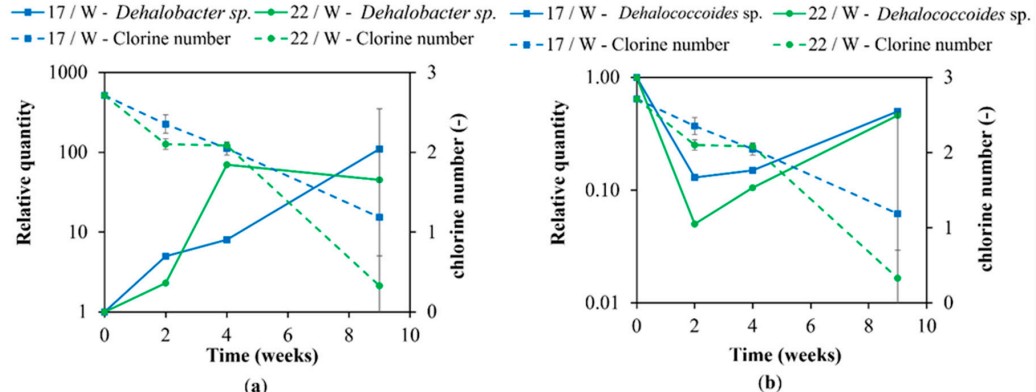

**Figure 8.** Correlation between (**a**) *Dehalobacter* sp. and (**b**) *Dehalococcoides* sp. and chlorine number at 17 and 22 °C with the addition of whey.

Similar trends of microbial communities were found either natural [27,71] or modified by the addition of carbon substrates [54,64]. In such complex microbial communities, many competitions can be found that are partially discussed above. Reductive dechlorination was found to be incomplete in sulfate-rich aquifers (sulfate concentrations greater than 400 mg/L) [72]. Mutual inhibition during concurrently reductive dechlorination of TCE and denitrification competing for electron donors was also found. Specific dechlorinated bacteria can be affected in different ways; thus, *Dehalococcoides* and *Dehalogenimonas* as obligate chlorine-respiring bacteria were inhibited by sulfate but induced by nitrate and facultative chlorine-respiring bacteria [73]. These competitions led to different consumptions of electron equivalents, typically to the detriment of chlorine-respiring bacteria. From the balance of the electron equivalent, the following order of electrons transferred resulted: reduction of $Fe^{3+}$ (1.22–1.91 mmol), sulfate (0.24–2.84 mmol), nitrate (0.375–0.75 mmol), and PCP reductive dechlorination (only 0.02 mmol) [62].

In general, symbiotic relationships in complex microbial communities appear to be essential for rapid and total biological reductive dechlorination [74]. Therefore, bacteria capable of fermentation, e.g., *Desulfovibrio* and *Clostridium* [22,75], play an important role in a dechlorinating consortium as electron donors through hydrogen production from introduced carbon sources [74]. *Clostridium butyricum* as a fermenting hydrogen-producing bacterium induced PCE dechlorination by *Dehalococcoides* spp. [52]. Symbiotic relationships can be found in all individual steps of gradual dechlorination. For example, during PCE dechlorination, members of the genus *Geobacter* provided transformation of PCE to dichloroethene and *Dehalococcoides* DCE to ethene [76]. *Sulfurospirillum multivorans* and *Dehalococcoides mccartyi* fully degraded PCE to ethene three times faster in mixed culture.

### 3.3. Principal Component Analysis

Principal component analyses (PCA) were completed for the main data on chlorinated ethenes, ethene, ethane concentrations, chlorine number, TOC, physical–chemical parameters and results of qPCR in two variants of laboratory tests—with addition of whey (Figure 9a) and without (Figure 9b). PCA captured 68.1 % of the total variance in the data with whey addition and 57.0 % total variance without whey addition. In both variants with and without addition of whey temperature, had a negative correlation with ORP and a positive correlation with $Fe^{2+}$ that indicated potential favorable reductive dechlorination conditions, iron-reducing conditions. Temperature had a strong negative correlation in both variants with cVOCs. TOC in the variant with whey had a negative correlation with all dechlorinating biomarkers, e.g., the bacteria *Dehalococcoides*, *Dehalobacter* and the dehalogenase genes *vcrA* and *bvcA* except for *Desulfitobacterium* spp. It is in compliance with a principle of biological reductive dechlorination, that substrate is degraded by bacteria that are able to utilize it and produce $H_2$ that is used by dechlorinating bacteria as an electron donor (Vogel a McCarty, 1985). It confirmed our findings that *Dehalococcoides* spp. started to grow in the ninth week of the test when TOC significantly decreased (see Figures 4 and 6). The growth of *Desulfitobacterium* spp. was most likely limited by other processes than by the degradation of TOC. Furthermore, PCA showed in both variants a positive correlation of temperature with chlorides (product of reductive dechlorination) and a negative correlation of temperature with chlorine number. In sum, the comprehensive statistical view of key results confirmed the findings and conclusions described and discussed above.

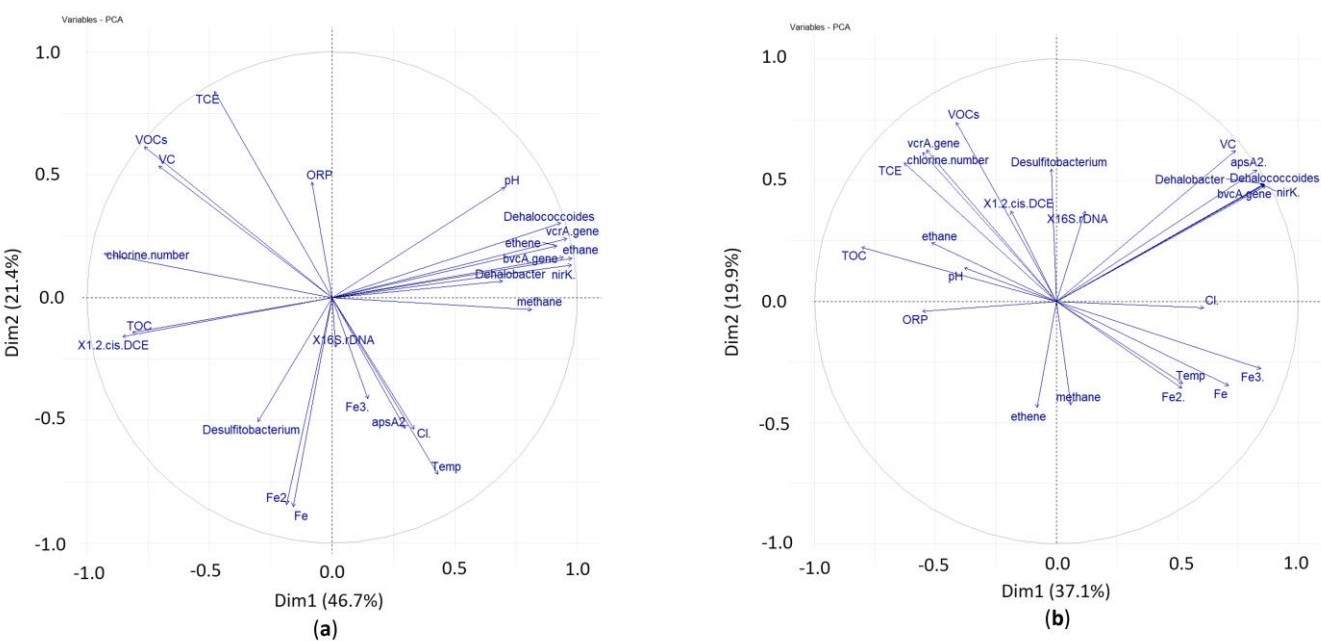

**Figure 9.** Principal component analysis of the qPCR, chemical and physical parameters, concentrations of iron, chlorides, chlorinated ethenes, ethene, and ethane. Refer to the text for the abbreviations used in test set-up (**a**) with whey and (**b**) without whey.

## 4. Conclusions

The laboratory experiments with contaminated groundwater and soil lasting nine weeks revealed trichloroethene dechlorination dependence on temperature (in the range of 12–22 °C) and on supply of substrate (whey) as energy, carbon, and sources of $H^+$ and $e^-$ sources. The qPCR results showed that the addition of whey caused a fundamental increase in total bacteria abundance as well as methanogens, nitrate, iron- and sulfate-reducing bacteria. Nevertheless, dechlorinating bacteria under study—*Dehalococcoides* spp., *Dehalobacter* spp., and *Desulfitobacterium* spp. can compete for energy and carbon sources and reducing equivalents. As evidence, complete biological reductive dechlorination

occurred in the variant with the addition of whey with incubation at 22 °C within nine weeks. The results indicated the temperature-promoted bioremediation of trichloroethene in combination with whey addition, as a cost-effective by-product, could be an effective variant of chlorinated ethenes degradation.

**Author Contributions:** All authors participate in conceptualizing the study, P.N. and V.K. undertook the material collection and sampling, J.S. and M.C. performed the molecular analyses, P.N. and R.Š. ensured all other analyses, J.N. and P.N. arranged resources and funding, P.N. and M.H. wrote the first draft of this study and all authors participated in reviewing and editing. All authors have read and agreed to the published version of the manuscript.

**Funding:** This research was funded by the Technology Agency of the Czech Republic, grant numbers TH01031225 and FW03010071.

**Institutional Review Board Statement:** Not applicable.

**Informed Consent Statement:** Not applicable.

**Data Availability Statement:** The data presented in this manuscript are available on request from the corresponding author.

**Conflicts of Interest:** The authors declare no conflict of interest.

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
