# Peer review of "Thermally Enhanced Biodegradation of TCE in Groundwater"

_water, doi:10.3390/w14213456_

Round 1
Reviewer 1 Report
Water, 2022, volume 14
Thermally Enhanced Bioremediation of TCE in Groundwater – Bench-Scale Testing
Review Comments:
First impressions
Is the research original, novel and important to the field?
- This work is original, novel and important to the field.
- The authors have done an excellent job in telling a complex story that involves analytical chemistry and qualitative and semiquantitative microbial community analysis.
Has the appropriate structure and language been used?
The structure is good and the language is appropriate.
Abstract
Is it really a summary?
- The abstract presents a good summary of the ideas and outcomes described in the paper.
- The authors have included keywords.
Does it include key findings?
Yes, the authors describe the key findings of their experiments.
Is it an appropriate length?
Yes, the abstract is of reasonable length.
Corrections:
Page 1, line 28-29 – “nine weeks from 2.5 to 0.1 at 22°C, to 1.1 at 17 °C and 1.7 at 12 °C” Insert the units (mg/L or mg/L)
Introduction
Is it effective, clear and well organized?
The authors have done a good job at stating the general problem of TCE degradation and their approach to dealing with this problem.
Does it really introduce and put into perspective what follows?
The authors have done a good job introducing the ideas of their study and how it fits into the larger field of groundwater remediation.
Suggest changes in organization and point authors to appropriate citations.
Overall, the introduction is very good.
Possible Improvement:
The introduction is a solid bit of writing that needs just one correction.
Corrections:
Page 2, lines 84-85 – “effective and economy temperature for total dechlorination”
Change to “effective and economically feasible temperature for total dechlorination”
Methodology
Can a colleague reproduce the experiments and get the same outcomes?
The methodology section describes each of the methods in depth such that the experiments could be reproduced by another researcher. The details of the experimental design are well written.
Did the authors include proper references to previously published methodology?
All of the sections dealing with different analytical methods were cited with historic and contemporary references.
Is the description of new methodology accurate? Yes
Corrections:
Page 3, lines 107-108 – “Furthermore, water had a relatively high concentration of total organic carbon increased in the groundwater tested by the addition of methanol” Change to “Furthermore, the water had a relatively high concentration of total organic carbon which increased in the groundwater tested by the addition of methanol”
Comment:
Page 5, line 164 – “whey was used as an organic substrate at a concentration of 1 g/l (100 g” Adding such a high concentration of available carbon could alter the microbial community structure and may impede TCE degradation and/or co-metabolism of TCE. In a future experiment, the authors may wish to add a carbon source that is two or three times the concentration of the chlorinated solvent.
Results and Discussion
Suggest improvements in the way data is shown
- The results and discussion sections are well written.
Comment on general logic and on justification of interpretations and conclusions
- The authors present a good case for the interpretations and conclusions based on the data generated from the various analytical methods.
Comment on the number of figures, tables and schemes
- The figures and tables are well-structured, concise and easy to read with the exception of Figure 9.
Write concisely and precisely which changes you recommend
- The results and discussion is well structured.
- The authors need to address the comments below.
Correction:
Page 6 lines 197-198 – “thus thermally enhanced anaerobic dechlorination could be costly effective” Change to “thus thermally enhanced anaerobic dechlorination could be costly, but effective”
Page 6, line 206 – “clogging of the pours caused by” Change to “clogging of the pores caused by”
Comment: Are the error bars in Figures 1 & 2 in Pages 6 & 7 based on triplicate analyses from one bottle or a single analysis from three separate bottles?
Figure 9 – It would be helpful if the authors increased the size of the font for the PC1 and PC2 axis so that the readers could see the percentage of variance captured by each PC. Did the authors consider doing any rotations of the data (e.g. equimax or varimax rotation) to see if there may arise a more distinct clustering of variables? It may be worthwhile to mention that the PCA captured 68.1% of the total variance in the data with whey addition and 57.0% total variance without whey addition.
List separately suggested changes in style, grammar and other small changes
- The writing style and grammar are excellent and need no improvement.
Suggest additional experiments or analyses
- There are no additional experiments or analyses needed for this manuscript.
- The experimental design, analytical methods and interpretation of the results are very good.
Make clear the need for changes/updates
- The quality of the science is and Technical English is excellent.
- The authors need to address just a few minor errors in the manuscript.
Ask yourself whether the manuscript should be published at all
- It is obvious that a great deal of effort went into this study.
- The quality of writing and science is excellent.
- Once the authors have addressed the comments mentioned above, then this manuscript will make for a great publication.
- The authors are to be commended for their interpretation of the complex relationship between microbial community structure, genetic expression and factors that influence the biodegradation of chlorinated pollutants.
Conclusion
Comment on importance, validity and generality of conclusions
- The conclusions are in line with the data generated by the authors.
Request toning down of unjustified claims and generalizations
- There are no unjustified claims nor any need to tone down the conclusions.
Request removal of redundancies and summaries
- There are no redundancies.
The abstract, not the conclusion, summarizes the study
- The abstract summarizes the study.
References, tables and figures
Check accuracy, number and citation appropriateness
- The citations are appropriate and accurate.
Comment on any footnotes
- No footnotes
Comment on figures, their quality and readability
- The figures are good and the use of color is essential in comprehending the analysis.
Assess completeness of legends, headers and axis labels
- The legends, headers and axes adequately describe the figures and tables.
Check presentation consistency
- There is a consistent level in the references and figures.
Comment on need for color in figures
- Due to the complexity of the figures, there is a definite need for color in the figures.
Author Response
Thank you for your valuable comments and suggestions. Please see the attachment.

Reviewer 2 Report
This paper investigates preparation of thermally enhanced bioremediation of TCE in groundwater bench-scale testing
· PCA analysis should be better explained.
· Figures of PCA are not clear, please make it more readable.
· Please check are all of your figures at least 300 dpi.
Author Response
First of all thank you for your comments and suggestions. Here are my answers.
Point 1: PCA analysis should be better explained.
Response 1: We changed whole paragraph 3.3 and tried to explain it better.
Point 2: Figures of PCA are not clear, please make it more readable.
Response 2: We changed axis labels and enlarged figures. Hopefully it is now more visible.
Point 3: Please check are all of your figures at least 300 dpi.
Response 3: I have checked it and all figures are 330 dpi, I hope it is sufficient.